# Preparation of Low-Cost Magnesium Oxychloride Cement Using Magnesium Residue Byproducts from the Production of Lithium Carbonate from Salt Lakes

**DOI:** 10.3390/ma14143899

**Published:** 2021-07-13

**Authors:** Pan Liu, Jinmei Dong, Chenggong Chang, Weixin Zheng, Xiuquan Liu, Xueying Xiao, Jing Wen

**Affiliations:** 1Key Laboratory of Comprehensive and Highly Efficient Utilization of Salt Lake Resources, Qinghai Institute of Salt Lake, Chinese Academy of Sciences, Xining 810008, China; lp8026@isl.ac.cn (P.L.); dongda839@isl.ac.cn (J.D.); ccg168@isl.ac.cn (C.C.); zhengweixin@isl.ac.cn (W.Z.); liuxiuquan19@mails.ucas.ac.cn (X.L.); 2Key Laboratory of Salt Lake Resources Chemistry of Qinghai Province, Xining 810008, China; 3University of Chinese Academy of Sciences, Beijing 100049, China

**Keywords:** magnesium oxychloride cement, magnesium residue, low-cost, calcination temperature, compressive strength

## Abstract

Magnesium oxychloride cement (abbreviated as MOC) was prepared using magnesium residue obtained from Li_2_CO_3_ extraction from salt lakes as raw material instead of light magnesium oxide. The properties of magnesium residue calcined at different temperatures were researched by XRD, SEM, LSPA, and SNAA. The preparation of MOC specimens with magnesium residue at different calcination temperatures (from 500 °C to 800 °C) and magnesium chloride solutions with different Baume degrees (24 Baume and 28 Baume) were studied. Compression strength tests were conducted at different curing ages from 3 d to 28 d. The hydration products, microstructure, and porosity of the specimens were analyzed by XRD, SEM, and MIP, respectively. The experimental results showed that magnesium residue’s properties, the BET surface gradually decreased and the crystal size increased with increasing calcination temperature, resulting in a longer setting time of MOC cement. Additionally, the experiment also indicated that magnesium chloride solution with a high Baume makes the MOC cement have higher strength. The MOC specimens prepared by magnesium residue at 800 °C and magnesium chloride solution Baume 28 exhibited a compressive of 123.3 MPa at 28 d, which met the mechanical property requirement of MOC materials. At the same time, magnesium oxychloride cement can be an effective alternative to Portland cement-based materials. In addition, it can reduce environmental pollution and improve the environmental impact of the construction industry, which is of great significance for sustainable development.

## 1. Introduction

Magnesium oxychloride cement, also known as magnesium oxychloride cementitious material, refers to a gas-hardening cementitious material formed by mixing a certain concentration of magnesium chloride solution with lightly burned oxide powder, which was first discovered by Sorel in 1867 [1,2,3]. Compared with ordinary Portland cement (abbreviated as OPC), MOC has the advantages of fast hardening, high strength, high cohesive force, high corrosion resistance, low thermal conductivity, and environmental protection [4,5,6], and the alkali of MOC. The degree is lower than OPC, so when used with glass fiber, it can be prepared into the cement with less corrosiveness [7]. Therefore, MOC has a wide range of applications in building materials and wood saving [8]. Compared to OPC, the calcination temperature of raw materials, the calcination temperature of MOC raw materials is only about 55% of OPC. At the same time, MOC cement can reduce carbon dioxide emissions. Hence, MOC materials have essential contributions to energy saving, capital saving, and emission reduction [9].

The traditional active magnesia used for MOC materials is calcined from magnesite at 700–900 °C [10,11]. China is one of the countries with the wealthiest magnesite resources in the world. The resources are rich, and the grade is high, but the distribution is shallow. Evenly, there are currently 27 proven mining areas. Liaoning has the most abundant magnesite reserves, accounting for 85.63% of the country [12,13]. The uneven distribution of magnesite resources dramatically limits MOC materials’ application in areas lacking magnesite [14,15]. At the same time, the salt lakes in Qinghai are widely distributed.

The extraction of potassium fertilizer in the salt lake area will produce many magnesium chloride waste liquid. The extraction of lithium carbonate products from the salt lake brine will also have many magnesium residues byproducts [16]. About 8–10 tons of magnesium chloride are produced for every 1 ton of potash fertilizer produced [17]. In 2017 alone, 3 million tons of magnesium chloride were produced [18]. Similarly, the magnesium residues produced in extracting lithium carbonate from the salt lakes several times that of lithium carbonate. Besides, the magnesium slag contains boron, and the separation of boron and magnesium is very difficult. Therefore, a large amount of magnesium slag cannot be used. If these magnesium residues are not used, they will waste resources and pollute the environment and directly affect the leading product, lithium carbonate [19]. The normal production of salt lakes has caused magnesium damage during the salt lake’s development [20]. The magnesium residues in extracting lithium carbonate must be treated or evaluated to reduce the so-called magnesium damage.

The mechanical properties of MOC cement depend mainly on the hydration rate and the types and relative content of hydration products [21]. In the last several decades, a lot of work has been conducted to study the microstructure, reaction mechanism, and strength development in the MgO-MgCl_2_-H_2_O ternary system formed by three reactants is, MgO, MgCl_2_ and H_2_O. Manufacturing MOC cement generally generates two major hydration phases produced in the ternary system are phase 3 (3Mg(OH)_2_·MgCl_2_·8H_2_O) and phase 5 (5Mg(OH)_2_·MgCl_2_·8H_2_O) at the temperature below100 °C [22,23]. Equations (1) and (2) are the theoretical formulas, which describe the reaction to generate phase 3 and phase 5 during hydration. Additionally, phase 5 appears shortly after mixing (about two hours after the paste is mixed). The mechanical interlocking and dense microstructure resulting from the crystals’ intergrowth are considered the main sources for the strength development of the MOC [24]. Hence, phase 5 is more preferable in the design of MOC. Crystal formation of these main reaction phase are controlled by several important factors, for instance magnesia reactivity [25], adopted magnesium oxide to magnesium chloride molar ratio (MgO/MgCl_2_) [26,27], water to magnesium chloride molar ratio (H_2_O/MgCl_2_) [28], physical admixture [29,30], chemical admixture [31,32] and curing conditions [33,34]. In the experiment, to obtain a dominance of phase 5 crystals in the MOC paste, it was found that a molar ratio of magnesium oxide to magnesium chloride between approximately 11 and 17 and a molar ratio of water to magnesium chloride between approximately 12 and 18 was the optimum choice [35].
(1)$$3\mathrm{MgO}+{\mathrm{MgCl}}_{2}+11H_{2}O\to 3{\mathrm{Mg}(\mathrm{OH})}_{2}\cdot {\mathrm{MgCl}}_{2}\cdot H_{2}O\;\left(\mathrm{Phase}\;3\right)$$
(2)$$5\mathrm{MgO}+{\mathrm{MgCl}}_{2}+13H_{2}O\to 5{\mathrm{Mg}(\mathrm{OH})}_{2}\cdot {\mathrm{MgCl}}_{2}\cdot 8H_{2}O\;\left(\mathrm{Phase}\;5\right)$$

In recent years, under the country’s implementation of green building materials and low-carbon environmental protection strategies, many researchers have paid close attention to the magnesium residue byproducts from the salt lakes. Peng Hao [36] used the byproduct brucite from the salt lake to extract potash fertilizer as the raw material. The calcination temperature was 550 °C, and the calcination time was 1 h to prepare active MgO. The magnesium oxychloride cement was prepared according to the active MgO:MgCl_2_:H_2_O ratio of 5:1:8. The maximum compressive strength measured under standard conditions for 28 d was 90 MPa. Zhang [37] investigated the use of salt lake magnesium residue to prepare high-purity magnesium hydroxide [Mg(OH)_2_] as fire-retardant material. Tan et al. [38] used the calcination method to extract the lithium byproduct and prepared magnesium phosphate cement with conforming setting time and mechanical properties without adding any retarder. Li et al. [39] improved the purity of MgO to 99.2% by improving experimental methods further. Wu et al. [40] used the calcination method to extract lithium byproducts to prepare magnesium oxysulfide cement and measured the setting time, mechanical properties, hydration products, and microstructure. The results showed that it is feasible to use the calcination method to extract lithium byproducts instead of light-burned powder to prepare magnesium oxysulfide cement. The production of high purity magnesium metal and magnesium hydroxide and magnesium oxide is an important application. However, these magnesium products’ production scale is very small and only uses magnesium residues, and a large amount of magnesium chloride waste liquid is not used. Therefore, it is imperative to study the preparation of magnesium oxychloride cement using salt lakes lithium-magnesium residue. At the same time, it is of great significance for the construction of resource-saving, environmentally friendly and sustainable development.

Therefore, in this study, magnesium residue calcined at various temperatures and magnesium chloride hexahydrate (MgCl_2_·6H_2_O) were used to prepare MOC cement. The influence of calcination temperature on active magnesia properties and its setting time, hydration heat, compressive strength, and hydration of obtained MOC cement was thoroughly researched.

## 2. Materials and Methods

### 2.1. Materials

The raw material of magnesium residue used in this experiment originated from the byproduct of Li_2_CO_3_ extracted by Qinghai Citic Guoan Technology Development Company. The magnesium residues’ elemental composition is tested by X-ray fluorescence (XRF, Axios PW4400) shown in Table 1. Figure 1 is the production process route of Li_2_CO_3_, including the production of magnesium residues. Figure 2 is the XRD pattern of magnesium residues. Moreover, the Rietveld method was employed to perform qualitative analysis [41] using Topas 4.2 software [42]. The results are shown in Table 2. It can be seen from the table that the main phases in the magnesium slag are MgO, Mg(OH)_2_, and Mg_3_B_2_O_6_; the content is 8%, 71%, 21%. Figure 3 also shows the EDS element diagram of the main elements in the magnesium residue. According to EDS, the following elements were present in the sample: Mg, O, B, C, Cl and Ca. These results confirmed the presence of phases detected by XRD, and were in agreement with data provided by XRF analysis. Qinghai Jiayoumeiye Ltd. (Qinghai, China) produced the bischofite used in the experiment, and its main component is MgCl_2_·6H_2_O. The chemical composition of magnesium chloride hexahydrate is shown in the literature [43].

### 2.2. Treatment Process of Magnesium Residues

The phase analysis of the magnesium residue in Figure 2 contains a large amount of Mg(OH)_2_. Combined with MOC’s hydration mechanism, the presence of Mg(OH)_2_ will have adverse effects on the formation of hydration products and MOC’s mechanical properties [44]. Therefore, before the preparation of MOC, the magnesium residue must be heat treated. Thermogravimetric analysis (TGA) and differential thermal scanning calorimetry (DSC) were used to analyze the magnesium residue. As shown in the TG/DSC curves in Figure 4, two endothermic peaks appear at about 100 °C and 364 °C. The first endothermic peaks at about 100 °C are related to the evaporation of free water in the magnesium residue. Additionally, the second endothermic peaks at about 364 °C is associated with the decomposition of Mg(OH)_2_ to MgO, as depicted in Equation (3). Therefore, in order to obtain activated magnesia, this experiment chooses to hold in a box-type resistance furnace at temperatures of 400 °C, 500 °C, 600 °C, 700 °C, and 800 °C for 1 h.
(3)$${\mathrm{Mg}(\mathrm{OH})}_{2}\to \mathrm{MgO}+H_{2}O$$

Because the magnesium residue’s size is different before calcination, to ensure the consistency of heat transfer during the calcination process, a ball mill is used to ball mill for 3 min and pass through a 120 mesh sieve before calcination. X-ray diffraction (XRD, PANalytical X’PROPert, Malvern panalytical, the Netherlands) was used to test the mineral composition of the magnesium residue calcined at various temperatures. CuKα radiation (λ = 0.15419 nm) was in the 2θ range from 5° to 70°. The Rietveld method was employed qualitative analysis suing Topas 4.2 software (BRUKER: Billerica, MA, USA) [42]. A scanning electron microscope (SEM, Hitachi/Oxford Instruments, Abingdon, UK) was used to observe the morphology of magnesium residue calcined at various temperatures. Besides, the particle size distribution before and after calcination was measured with a laser particle size analyzer (LSPA, Mastersizer 2000, Malvern, UK). The static nitrogen adsorption analyzer (JW-BK 112, Canta Instruments Inc, USA) was used to analyze the specific surface area and porosity of the magnesium slag calcined at different temperatures.

### 2.3. Specimen Preparation

In this experiment, magnesium oxide’s molar ratio to magnesium chloride is 8.5, and the Baume degree of magnesium chloride solution is 24 and 28, respectively. After mixing according to the test ratio, stir at low speed by hand for 30 s, stir at high speed for 60 s, and then stir and mix. The uniform MOC paste is then cast into 2 × 2 × 2 cm^3^ steel molds and cure at a temperature of 20 ± 2 °C and relative humidity of 60% ± 5%. During the filling process, a steel pestle was used to expel the bubbles in the MOC specimen, and the surface of the MOC specimen was flattened. Demoulding after 1 d curing at room temperature, continue curing to 3, 7, and 28 d of age and determine the compressive strength of the MOC specimens.

### 2.4. Analysis Method

#### 2.4.1. Setting Time

The setting time plays an important role in the actual production of MOC cement. The specific test method of MOC cement should refer to GB/T 1346-2001 [45].

#### 2.4.2. Compressive Strength Analysis

The compressive strength of a hardened MOC specimen is measured by HYE-300B-D MEMS hydraulic servo pressure testing machine. The compressive strength test is performed at ages of 3, 7, and 28 d. The specific method of compressive strength refers to the Chinese National Standard GB/T17671-1999 [46].

#### 2.4.3. Crystalline Phase and Microstructure

The X-ray diffractometer (XRD) used was the X’pert Pro diffractometer produced by PANalytical Corporation in the Netherlands for phase analysis. The test conditions: Cu/Kα, tube pressure 40 kV, tube flow 30 mA, scanning range (2θ) is 5~70°.At the same time, Topas 4.2 software is used for phase quantitative analysis and was used for full-spectrum fitting of the XRD spectrum. The Rietveld method is used for quantitative analysis of the content of each phase.

The JSM-5610LV low vacuum scanning electron microscope produced by Japan Electron Optics Corporation (JEOL) was used to observe the sample’s microstructure and morphology. The acceleration voltage of the SEM is 20 kV, and the resolution is 3.5 nm. Take the MOC samples of the corresponding age to make thin and flat fragments, paste the samples on the copper sample holder with conductive glue, and use them for SEM analysis after vacuum plating.

#### 2.4.4. Pore Structure Test

The AutoPore IV9500 mercury porosimeter produced by Mike Instrument Co., Ltd. (Moscow, Russia). was used to analyze the sample’s pore structure; the pressure test range is 0~3.0 × 10 Pa, the detection diameter of the sample is 5~3,400,000 nm, and the accuracy is 0.11%. The sample size’s test conditions are: the sample size is 2 to 3 g; the sample size does not exceed 15 × 15 × 15 mm^3^, and the sample is vacuum dried before testing.

#### 2.4.5. Determination of Hydration Heat

Reference methods measured the hydration heat and hydration rate of MOC, and the eight-channel microcalorimeter (TAM) was used for testing. Method: The magnesium oxide powder was weighed and sealed in an ampere bottle. The prepared magnesium chloride solution was inhaled into the syringe and then put into an eight-channel calorimeter. After the baseline on display is stable, push the syringe to mix magnesium chloride solution with magnesium oxide powder, start the stirring motor, make the two thoroughly mixed and fully hydrated, and stop the test after waiting for the heat release rate curve on display to return to the baseline.

## 3. Results and Discussion

### 3.1. Characterization of Magnesia by Calcination at Different Temperatures Chemical Composition

Figure 5 shows the X-ray diffraction spectra of the magnesium residues after calcination at different temperatures. It can be seen from the figure the main mineral phases present in magnesium residue are MgO, Mg(OH)_2,_ and Mg_3_B_2_O_6_ before calcination. The diffraction peaks of Mg(OH)_2_ disappeared gradually, while the intensity of the diffraction peak of MgO increased with increasing calcination temperature. Moreover, Topas 4.2 software was used to analyze the number of the main crystal phases are listed in Table 3. The results shows that the relative content of MgO gradually increased, and the relative content of Mg(OH)_2_ gradually decreased with increasing temperature. The data in Table 4 shows that the contents of Mg(OH)_2_ and MgO in magnesium residue are 43.32% and 39.04% before calcination, respectively. After calcination, Mg(OH)_2_ decomposed completely when the calcination temperature is higher than 600 °C, whereas the MgO content is significantly increased up to 82.89% for the calcination temperature of 800 °C. Additionally, the magnesia derived from the magnesium residue contained ~20% insoluble Mg_3_B_2_O_6_ and a small amount of NaCl (close to 1%).

The type of hydration products and hydration rate of MgO in the magnesium chloride solution relate not only to the molar ratio of a-MgO/MgCl_2_ but also to the extent of reactivity of magnesia with water in MOC cement formulations [40]. The different types of magnesia derived from the magnesium residues are shown in Table 3. It can be seen from Table 4, the crystal size gradually increased, and the BET surface decreased with increasing calcination temperature. The results show that the crystallinity of magnesium oxide increases as the calcination temperature increases, and the magnesia particle densified at high calcination temperature. Further, when the calcination temperature increased from 400 to 800 °C, the citric color changing time was extended from 51 to 196 s, which indicates that less crystalline magnesium oxide reacted faster with water. In addition, the activity of magnesium oxide also increases with the increase in the calcination temperature. For example, the activity is 56.6% at 400 °C, and the activity is 70.0% at 800 °C. Therefore, it can be speculated that the setting and hardening of MOC cement were influenced by the calcination temperature [47].

### 3.2. Micromorphology

SEM images of magnesium residue before and after calcined at different temperatures are shown in Figure 6, which reflects the radiation tendency of the cumulative volume fraction of particles of magnesium residue at different calcination temperatures. As present in Figure 6a, magnesium residue’s surface before calcination is covered by a large number of flake crystals with diameters of approximately 6μm. With the increase of calcination temperatures, flake crystals disappear. As seen in Figure 6b, some small particles’ agglomeration appears when the calcination temperature is low (400 °C). This is due to the high BET surface area of magnesium residue (Table 4). Compared with Figure 6e,f, after calcination at 700 and 800 °C, fine particles are aggregated, and therefore, bulk particles were obtained. It appears that the largest size of the bulk particles is approximately 8 μm, calcined at 800 °C (Figure 6f). Otherwise, calcined at 700 °C, BET surface area is larger and crystallization degree is lower, surface defects are more, resulting in higher reactivity. This is consistent with the MgO activity data from hydrological test.

### 3.3. Particle Size Distribution

Figure 7 shows the particle size distribution curve and particle size distribution diagram of the magnesium residue at various calcination temperatures. It can be seen from the particle size distribution diagram that the particle size distribution range of the uncalcined sample is relatively wide. With the increase of the calcination temperature, the particle size distribution first narrows and then widens. This is because the uncalcined sample contains a large amount of Mg(OH)_2_. As the calcination temperature increases, the Mg(OH)_2_ decomposition particle size distribution narrows. When the calcination temperature increases, the small particles of MgO gradually aggregate into large particles, which is consistent with the SEM results.

The cumulative particle size distribution of magnesium residue at different calcination temperatures is shown in Figure 7b. It can be seen that with increase of calcination temperature, the particles are less fine, more coarse, and the average particle size increases. For example, the content of magnesium residue with particle size of less than 60 μm is 76.60% before calcination, whereas the calcination temperature increases from 400 °C to 800 °C, the content of magnesium residue decreases from 84.51% to 82.28%.

### 3.4. Setting Time of MOC

The setting time of magnesium oxychloride cement includes the initial setting and final setting time, closely related to change in cement’s rheological properties after mixing MgO with MgCl_2_ solution.

Figure 8 shows the setting time of MOC cement prepared by calcining magnesium residue at different temperatures. It can be seen that the setting time of the MOC cement increased with increasing calcination temperature. For instance, When the calcination temperature is 500 °C and the Baume degree of magnesium chloride is 24, the final setting time is only 3.9 h. When the calcination temperature is 800 °C, the final setting time is 46 h. This is associated with the BET surface of crystal, as discussed in Section 3.1. Therefore, highly crystallization and low BET surface magnesia contained few active sites restricting reactions with Mg^2+^, OH^−^, H_2_O, Cl^−^ in Magnesium chloride solution. This delayed the formation of hydration products and extended the setting time of MOC. The heat-release-rate curves for the MOC pastes with magnesium residue calcined at different temperatures are presented in Figure 9. It can be seen that the induction periods and acceleration period became longer with calcination temperature, which can also prove the effects laws of calcination temperature of magnesia on setting the time on MOC cement.

As shown in Figure 8, when the Baume degree of the magnesium chloride solution increases to 28, the MOC setting becomes faster. When the Baume degree of the magnesium chloride solution rises from 24 to 28, and the calcination temperature is 800 °C, the initial setting and final setting time are reduced by 6 h and 5 h, respectively. Thence, MOC cement prepared with magnesium residue calcined at lower temperature and increase Baume degree of magnesium chloride solution improve hardening characteristics.

### 3.5. Compressive Strength of MOC

MOC pastes’ compressive strength with magnesium residue calcined at different temperatures is measured at curing ages of 3 d, 7 d, and 28 d (Figure 10). As mentioned, the compressive strength of MOC relates to the molar ratio of active magnesium oxide to magnesium chloride [48,49], hydration rate [50] and hydration phase composition [51]. Therefore, it is of great significance to study the influence of different calcination temperatures and different Baume solutions on MOC’s compressive strength. At a constant curing age, the MOC specimens’ compressive strengths with high Baume prepared by magnesium residue at a constant calcination temperature are higher than those with low Baume. For example, the compressive strength of specimens prepared by magnesium residue calcined at 600 °C with Baume 28 is 102.0 MPa at 28 d, which is 100.4% higher than this of the specimen Baume 24. At the same Baume, the compressive strengths of MOC specimens prepared by magnesium residue calcined 800 °C with Baume 28 are 123.3 MPa at 28 d, which is 95.3% higher than this is a specimen with 500 °C.

### 3.6. Effect of the Magnesium Residue Calcination Temperature on the Hydration Products and Microstructure

Figure 11 shows the XRD patterns of MOC specimens prepared from magnesium residue at different calcination temperatures and Baume degrees after curing in the air for 28 d. The XRD spectra that the main hydration product in MOC was 5·1·8 phase (The main intensity phase of MOC), Mg(OH)_2,_ and MgO, Mg_3_B_2_O_5_ phase was also found in the MOC paste. Topas 4.2 software was used to analyze each phase quantitatively, and the results are shown in Table 5. It can be observed that MOC prepared with magnesium residue calcined at 800 °C and 28 Baume had the highest relative quantity of 5·1·8 phase (82.14%) after curing for 28 d. This is consistent with our findings regarding the development of compressive strength.

Figure 12 28-day SEM morphology of MOC specimens prepared by calcining magnesium residue at different temperatures. The figure shown that when the calcination temperature is 600 °C, there are more rod-like crystals in the MOC specimen. Still, the morphology of the needle-rod-like crystals is relatively irregular. Although the rod-shaped crystals overlap each other to form a network structure, the network structure is somewhat loose. When the calcination temperature is 800 °C there are more gel-like crystals in the MOC specimens. A large number of gel-like crystals increase the MOC specimens’ density, thereby enhancing the compressive performance of the MOC specimens. In summary, as the calcination temperature increases, the compactness of the MOC specimen is gradually improved. The felt-like structure formed by the mutual filling of rod-like crystals and gel-like crystals, or the structure with gel-like crystals as the main body, has relatively improved compactness, thereby improving the compression resistance of the MOC specimen.

In Figure 12, 28-day SEM morphology of MOC specimens are prepared by magnesium residue at different calcination temperatures with 28 Baume degrees. It can be seen from the figure that when the Baume degree of the MgCl_2_ solution is 28 and the calcination temperature is 600 °C, the finely divided gel-like crystals are stacked on top of each other and woven into a sparse and dense network structure but compared to the MOC when the Baume degree is 24 The density of the test piece has a certain degree of improvement. When the calcination temperature is 800 °C when the MgCl_2_ solution is 28 Baume, there are more gelatinous crystals in the MOC specimens. The gelatinous crystals are stacked on each other than when the calcination temperature is 800 °C when the MgCl_2_ solution is 24 Baume. The compactness of the parts is further improved.

In summary, as the Baume degree of MgCl_2_ solution increases, the density of MOC specimens gradually increases. When preparing MOC specimens with a fixed molar ratio, increasing the Baume degree of MgCl_2_ solution reduces the free water content in MOC specimens. The free water in the test piece evaporates continuously, introducing more voids in the test piece. Therefore, when the Baume degree of the MgCl_2_ solution is low, more free water vaporizes in the MOC specimen, and the structure of the MOC specimen is looser and porous; when the Baume degree of the MgCl_2_ solution is large, the free water in the MOC vaporizes out Less, the overall structure of the specimen is denser.

### 3.7. Effect of the Magnesium Residue Calcination Temperature on MOC Porosity

The mechanical strength of MOC cement is related to the content of crystals and microstructure of the hydration products, porosity and pore distribution [52,53]. As shown in Figure 13, the cumulative porosity of MOC cement with increasing calcination temperature and Baume. For example, the cumulative porosities of MOC cement prepared with magnesium residue calcined at and 800 °C were 12.78% and 6.72%, respectively, at 24 Baume. Additionally, the cumulative porosities of MOC cement prepared with magnesium residue calcined at 600 °C and 800 °C were 10.65% and 6.04%, respectively, at 28 Baume. The degree of crystallization in magnesium residue increased, and crystal defects were minimized when magnesium residue was calcined at high temperatures. The pores in the MgO particles and the pores between magnesium residue particles decreased. When the Baume degree increases, the reaction becomes more complete, increasing the amount of hydration products (Table 5 Topas 4.2), thus having higher compressive strength. The higher degree of hydration limited pores growth between particles and hydration products and pores between hydration products.

## 4. Conclusions

MOC cement was prepared with magnesium residue from the produced Li_2_CO_3_ from salt lakes in this study. According to the experimental results obtained, the following general conclusion can be drawn:After different temperatures calcination, the main phase MgO content increases, and the Mg_3_B_2_O_6_ content are almost unchanged, whereas the Mg(OH)_2_ phase disappeared.The results show that with the increase of calcining temperature, the crystallization degree of magnesium residues increase, BET surface area decreased and reactivity with water of the calcined magnesium residue increased with increasing the calcination temperature. Therefore, with increased calcination temperature of magnesium residues, the setting time of the MOC cement is prolonged.The Baume degree of the magnesium chloride solution has an essential influence on MOC cement’s compressive strength. The calcination temperature is 800 °C, the molar ratio of magnesium oxide to magnesium chloride is 8.5, and the Baume degree of the magnesium chloride solution is 28, the compressive strength of MOC can reach 123.3 MPa after 28 d.Taking into account the regional characteristics of Qinghai, the use of the byproduct of extracting lithium carbonate from salt lakes to prepare MOC cement can save resources, protect the environment, and reduce the production cost of MOC, which is of great significance for the industrial production of MOC cement and the expansion of the application fields of MOC materials.Finally, it is not clear or not impurities (main Mg_3_B_2_O_6_) influence crystal defects and MOC formation. This requires further study. It can be studied by nanoindentation, atomic force microscopy, molecular dynamics simulation, and DFT calculation methods.

## Figures and Tables

**Figure 1 materials-14-03899-f001:**
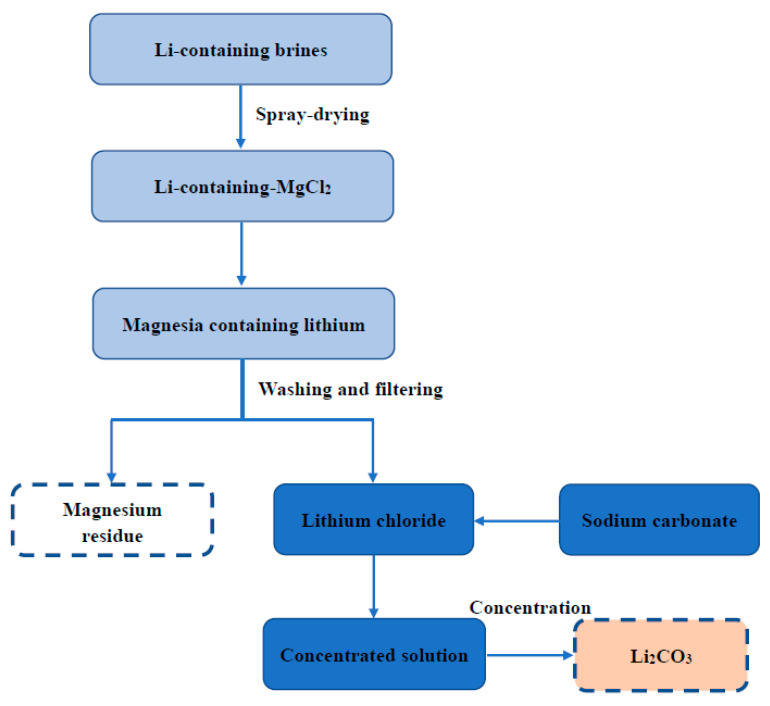
The technology route of Li_2_CO_3_ production by calcination from salt lake.

**Figure 2 materials-14-03899-f002:**
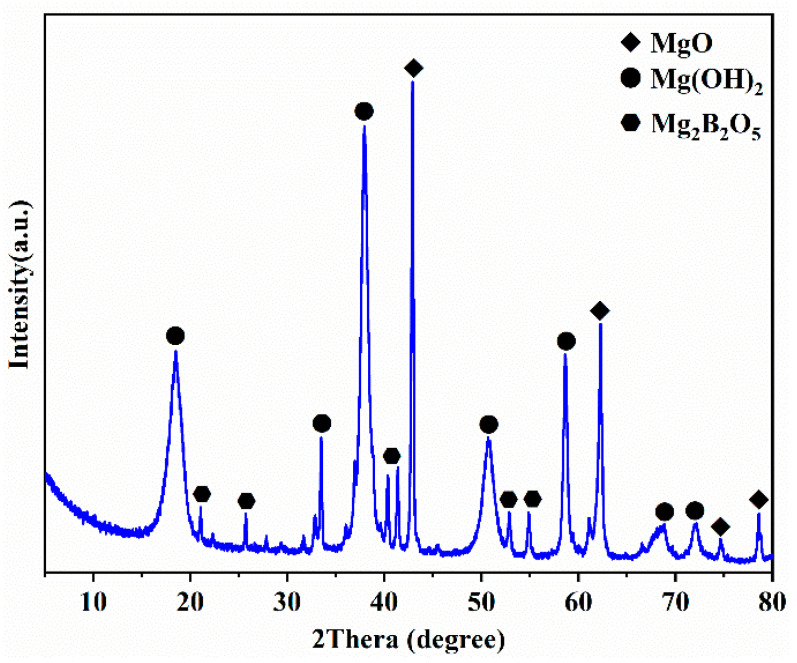
XRD pattern of magnesium residue before calcination.

**Figure 3 materials-14-03899-f003:**
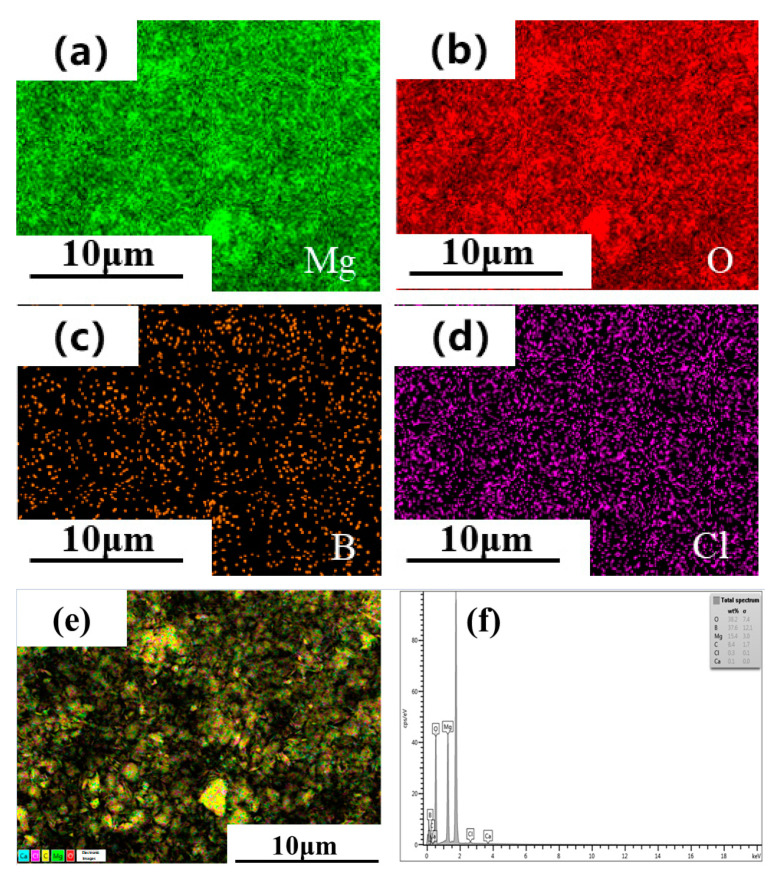
EDS spectrum of magnesium residue: (**a**) Mg; (**b**) O; (**c**) B; (**d**) CI; (**e**) EDS elemental maps of major elements; (**f**) EDS spectrum.

**Figure 4 materials-14-03899-f004:**
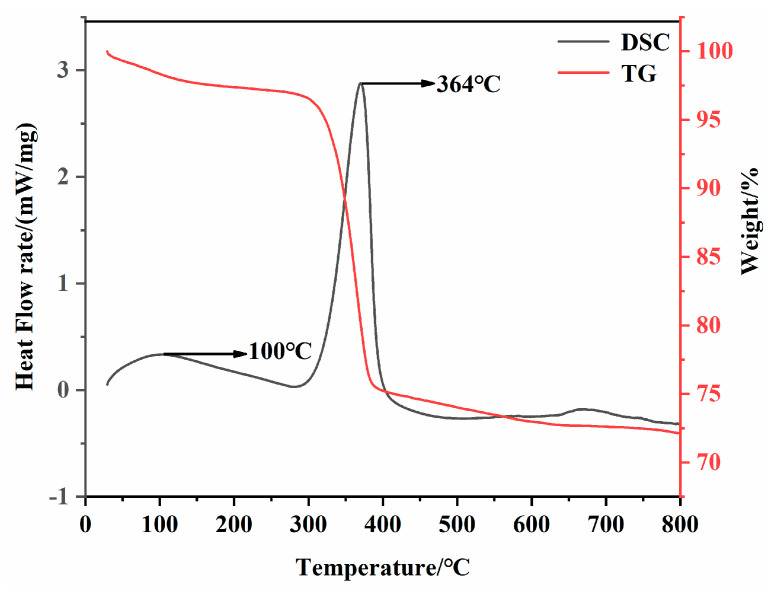
TG-DSC of magnesium residue before calcination.

**Figure 5 materials-14-03899-f005:**
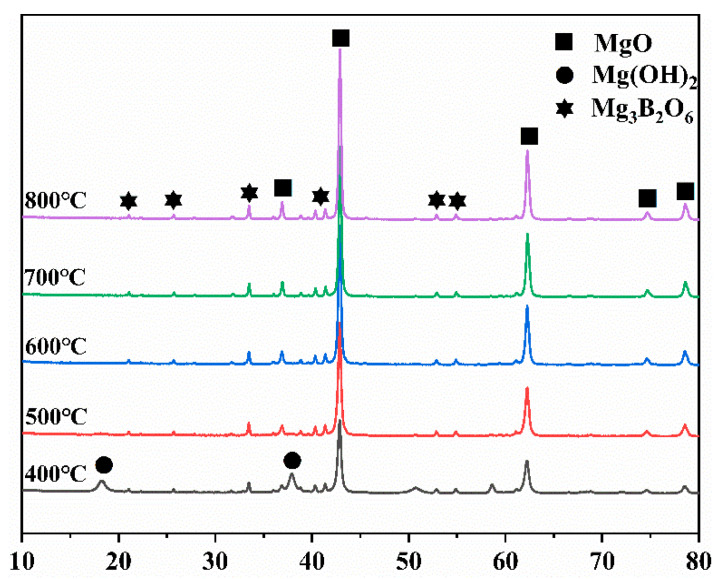
XRD patterns of magnesium residue calcined at different temperatures.

**Figure 6 materials-14-03899-f006:**
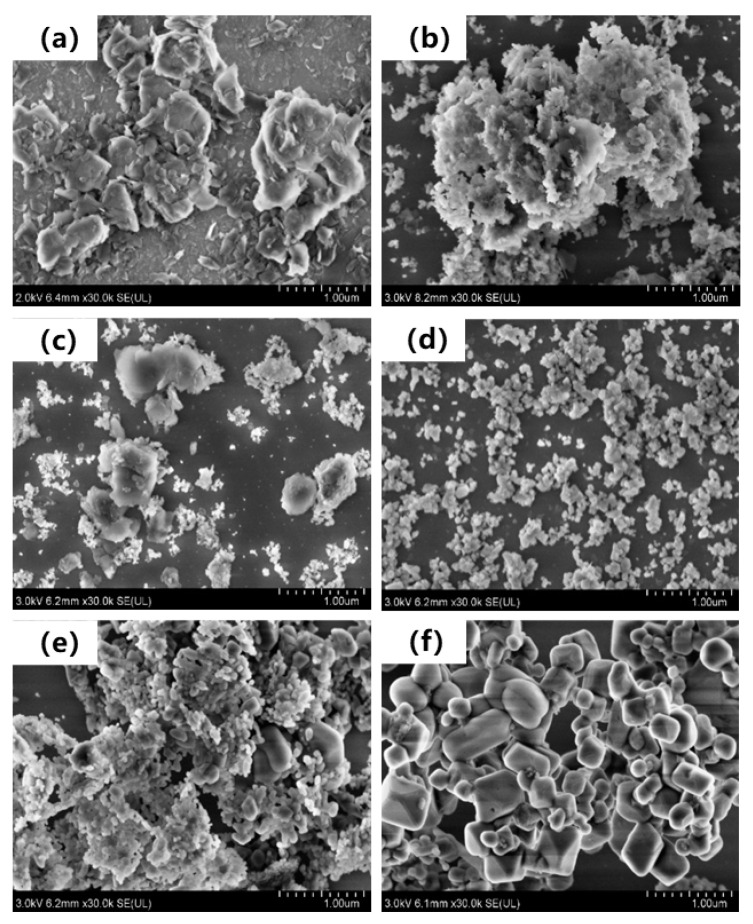
SEM images of the magnesium residue calcined at different temperatures. (**a**) 0 °C. (**b**) 400 °C (**c**) 500 °C (**d**) 600 °C (**e**) 700 °C (**f**) 800 °C.

**Figure 7 materials-14-03899-f007:**
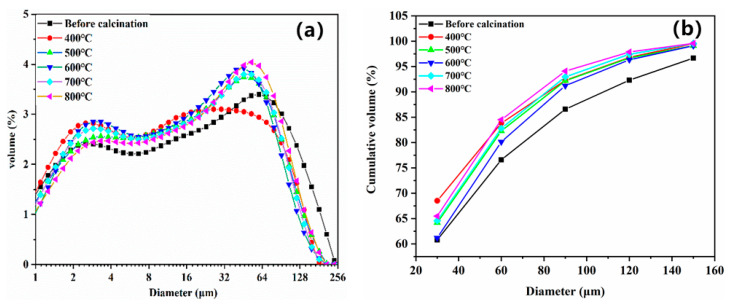
Particle size distribution of magnesium residue at different calcination temperatures: (**a**) particle size distribution, (**b**) cumulative particle size distribution.

**Figure 8 materials-14-03899-f008:**
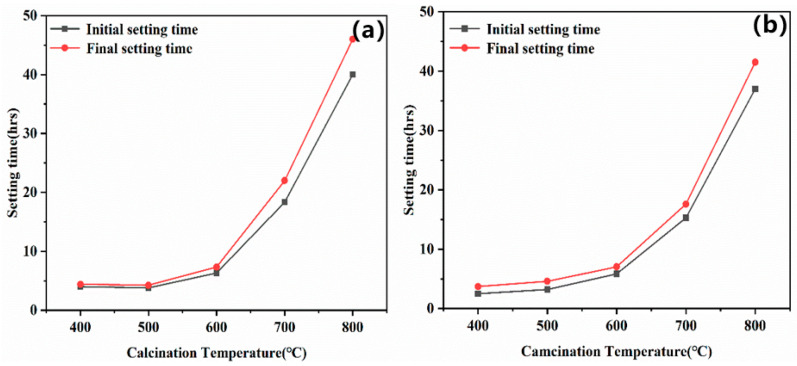
Effect of the calcination temperature applied to obtain active magnesia on the setting time of MOC prepared with Baume degree of magnesium chloride solution equal to 24 (**a**) and 28 (**b**).

**Figure 9 materials-14-03899-f009:**
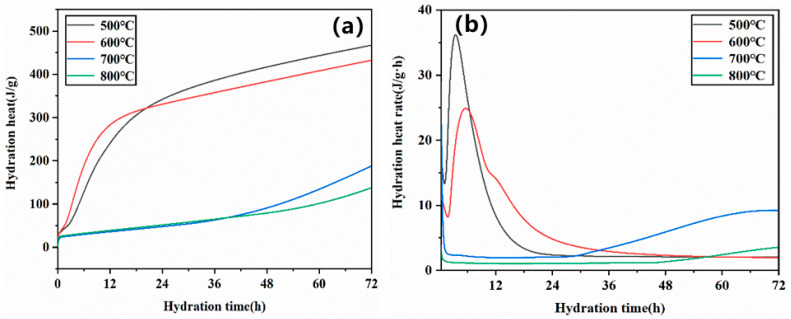
Hydration exothermic curves of different type of MOC cement: (**a**) Hydration heat (**b**) Hydration reaction rate.

**Figure 10 materials-14-03899-f010:**
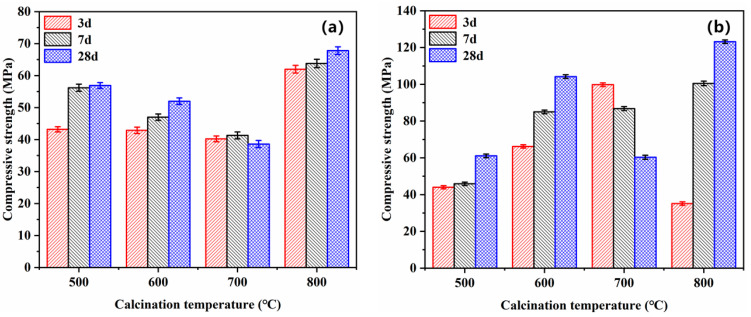
Effect of Baume and the calcination temperature of magnesium residue on the compressive strengths of MOC: (**a**) 24 Baume (**b**) 28 Baume.

**Figure 11 materials-14-03899-f011:**
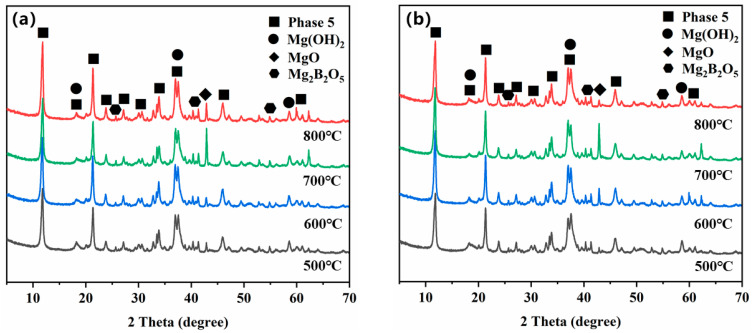
X-ray diffractograms of the MOC paste at different Baume and different calcination temperature of magnesium residue for 28 d: (**a**) 24 Baume (**b**) 28 Baume.

**Figure 12 materials-14-03899-f012:**
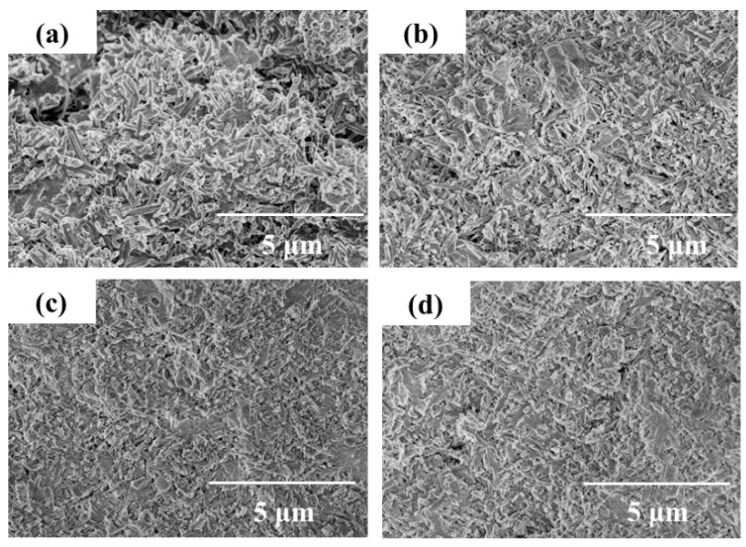
SEM topography of 28-dayMOC specimens at different temperatures and Baume degrees: (**a**) 600 °C, 24 Baume (**b**) 600 °C, 28 Baume (**c**) 800 °C, 24 Baume (**d**) 800 °C, 28 Baume.

**Figure 13 materials-14-03899-f013:**
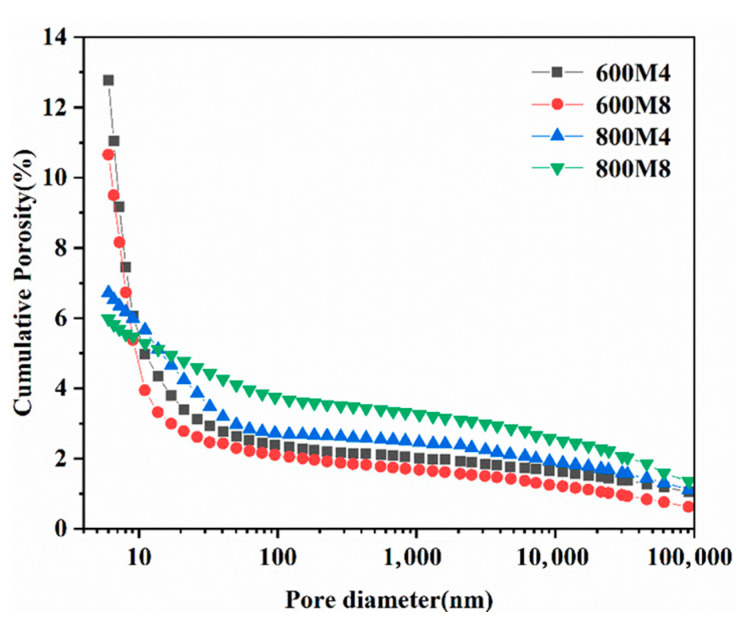
Cumulative porosities of MOC with magnesium residue calcined at different temperatures and Baume after hydration for 28 d. (In 800M4, 800 represents the calcination temperature of 800 °C, M4 represents the Baume degree of magnesium chloride is 24, and the M8d love table represents the Baume degree of magnesium chloride is 28).

**Table 1 materials-14-03899-t001:** The elemental composition of the magnesium residues.

Composition	MgO	B_2_O_3_	Na_2_O	SiO_2_	Li_2_O	CaO	K_2_O	SO_3_	Al_2_O_3_	LOI
Content/%	81.36	4.58	1.32	0.03	0.86	0.81	0.096	0.27	0.055	10.62

**Table 2 materials-14-03899-t002:** XRD quantitative analysis of magnesium residues.

Composition	MgO	Mg(OH)_2_	Mg_3_B_2_O_6_
Content/%	8	71	21

**Table 3 materials-14-03899-t003:** Mineral phase composition of the magnesium residue calcined at different temperatures.

Temperature/°C	MgO	Mg(OH)_2_	Mg_3_B_2_O_6_	NaCl	Crystallite Dimensions/nm	R
0	39.04	42.32	18.08	0.56	66.2	12.36
400	65.53	15.48	18.59	0.40	29.8	13.59
500	79.04	1.60	18.80	0.55	34.8	8.93
600	80.92	0	18.50	0.58	45.6	9.27
700	83.10	0	16.05	0.84	52.7	9.16
800	82.89	0	16.17	0.94	62.2	10.58

**Table 4 materials-14-03899-t004:** Properties of magnesia obtained from the magnesium residue calcined at different temperature.

Temperature/°C	Citric Acid Color-Changing Time/s	Active Magnesia Content/%	BET Surface/(m^2^/g)
400	51	56.6	27.8
500	73	68.4	27.5
600	94	71.3	26.3
700	151	73.0	18.6
800	196	70.0	8.9

**Table 5 materials-14-03899-t005:** The phase and their quantities in MOC with different Baume and different calcination temperature of magnesium residue for 28 d.

Temperature/°C	Baume	5·1·8 Phase	Mg(OH)_2_	MgO	Mg_2_B_2_O_5_	Rwp/%
500	24	76.53	8.79	2.40	12.28	10.05
	28	70.81	7.72	9.48	11.98	10.40
600	24	78.42	5.87	4.57	11.15	10.34
	28	78.43	4.57	5.76	11.15	10.32
700	24	77.73	8.37	3.12	10.77	10.44
	28	67.40	9.65	9.93	11.98	10.22
800	24	79.32	4.62	5.83	10.23	10.23
	28	82.14	2.54	4.46	10.86	10.34

## Data Availability

The data presented in this study are available on request from the corresponding authors.

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
