# Peer review of "Preparation of Low-Cost Magnesium Oxychloride Cement Using Magnesium Residue Byproducts from the Production of Lithium Carbonate from Salt Lakes"

_materials, 2021, doi:10.3390/ma14143899_

Round 1

Reviewer 1 Report

Manuscryp is very interesting. In my opinion, the presented results are very valuable because magnesium oxychloride cement can be an effective alternative to Portland cement-based materials. It is very important due to the implementation of the main assumptions of the sustainable development policy. 

The logical build and structure of the manuscript is very good.The technical English is also good The manuscript is concise and well organized.  The authors carefully prepared the figures. the manuscript  pointed out sufficiently what this work brings really as new in terms of progress in comparison to the works already done. 

Author Response

REVIEWER REPORT(S):

Referee:1

Comments and Suggestions for Authors:

Manuscript is very interesting. In my opinion, the presented results are very valuable because magnesium oxychloride cement can be an effective alternative to Portland cement-based materials. It is very important due to the implementation of the main assumptions of the sustainable development policy.The logical build and structure of the manuscript is very good. The technical English is also good The manuscript is concise and well organized. The authors carefully prepared the figures. the manuscript pointed out sufficiently what this work brings really as new in terms of progress in comparison to the works already done.

Author reply: Thank you very much for your affirmation of the manuscript.

Reviewer 2 Report

The work is very valuable and the logo has a broad spectrum of research. It addresses the issue and is introduced with environmental protection and sustainable seasoning.

Comments:

Check the space between citation [x] and text throughout the article.

There are not enough places in many places. Incorrect table references on pages 7 and 8. Refer to table 5.

Capital letter in the title of point 3.4. and 3.5. There is no information in the research of the research center on how many are tested.

Throughout the article and figures, please standardize the units of units, mark them with spaces or without.

Double numbering on the literature list. 

Author Response

Comment 1: Check the space between citation [x] and text throughout the article.

Author reply: Thanks for your helpful remind, It has been carefully checked the space between citation [x] and text throughout the article.

Comment 2: There are not enough places in many places. Incorrect table references on pages 7 and 8. Refer to table 5.Capital letter in the title of point 3.4.and 3.5.There is no information in the research of the research center on how many are tested. Throughout the article and figures, please standardize the units of units, mark them with spaces or without. Double numbering on the literature list.

Author reply: Thanks for your kindly remined again, issues in the manuscript have been corrected.

Reviewer 3 Report

The development of alternative binders for Portland cement by utilizing industrial by-products has long been a popular research topic. Although the use of magnesium oxychloride cement (MOC) is also a promising strategy, it has been relatively less known. In this context, this study dealing with the characterization of MOC with the calcination temperature of magnesium residue as the main variable was done in a timely manner. However, the current manuscript needs to be revised to be accepted.

Comments

First of all, why aren't there line numbers in the manuscript? (This makes it difficult to write comments).

Abstract: Define the abbreviation, MOC.

First paragraph of the introduction: Define the MOC, again. And, in addition to the calcination temperature, highlight the advantages of MOC over OPC in terms of CO2 emissions.

Between the 3rd and 4th paragraphs of the introduction: The connection between the two paragraphs is not natural. In particular, there is not enough explanation on the justification for using MgCl2 liquid for MOC manufacturing. In addition, is MgCl2 liquid harmful to the human body? (Like strong alkali activators used in geopolymers).

Equations (1) to (3): The arrows are missing. Check it.

Caption of Figure 1: Li2CO3 is correct.

Table 1: The sum of contents is not 100%. What's the rest? In addition, the ion content cannot be derived from the XRF analysis. Only the oxide composition can be determined by this analysis. Correct it.

All figures: change the shape of the marker. Circle and hexagon are not clearly distinguished.

Bottom part of page 5: The λ value is probably wrong. Check it carefully. Also, move the sentences about the equipment and methods used for XRD analysis to Section 2.1.

Figure 4: As a result of TG analysis, dehydration of Mg(OH)2 was almost completed at 400°C. Nevertheless, why did the authors set the firing temperature between 400-800°C as the main test variable?

Section 2.3: Add a new table showing mix proportions for better understanding.

Top of page 7: The sentences describing the procedures of XRD and SEM analyses should be moved to where they first appear.

Bottom of page 7: How was the crystal size determined?

Figure 5: As confirmed by the TG analysis, the dehydration of Mg(OH)2 was almost completed at 400°C. Nevertheless, why was the Mg(OH)2 peak detected in the 400°C sample?

Table 5: The BET surface area of the 800°C sample is remarkably low. Didn't the authors consider the possibility of melting as a cause?

Middle part of page 10: I don't agree with the sentence “This is associated with the degree of crystallization”, because no evidence is confirmed in the XRD patterns. If there was a change in the degree of crystallization, the change from the amorphous hump to the crystallization peak should have been confirmed with increasing temperature. Based on the SEM images and BET measurement results, I would like to suggest the reduction of the specific surface area as a more probable cause regardless of the change in the degree of crystallization. Consider this, and if the authors agree with me, revise the relevant sentences.

Page 11: "As mentioned" is better than "As we all know"

Author Response

Referee:3

Comment 1: First of all, why aren't there line numbers in the manuscript? (This makes it difficult to write comments).

Author reply: I'm really sorry that I forgot to add the line numbers in the manuscript due to my negligence.

Comment 2: Abstract: Define the abbreviation, MOC. First paragraph of the introduction: Define the MOC, again. And, in addition to the calcination temperature, highlight the advantages of MOC over OPC in terms of CO2 emissions.

Author reply: The author has made relevant revisions in the original manuscript.

Comment 3: Between the 3rd and 4th paragraphs of the introduction: The connection between the two paragraphs is not natural. In particular, there is not enough explanation on the justification for using MgCl2 liquid for MOC manufacturing. In addition, is MgCl2 liquid harmful to the human body? (Like strong alkali activators used in geopolymers).

Author reply: The salt lake produces 3 million tons of magnesium chloride as a by-product each year in the process of potash fertilizer production, which causes great harm to the environment. However, the amount of magnesium oxychloride used to prepare other products is very small, and the amount of magnesium chloride used to prepare MOC cement is large and does not need to be refined. Magnesium chloride is harmless in small amounts and can cause harm in excess of a certain amount.

Comment 4: Equations (1) to (3): The arrows are missing. Check it. Caption of Figure 1: Li2CO3 is correct.

Author reply: The author has made relevant revisions in the original manuscript.

Comment 5: Table 1: The sum of contents is not 100%. What's the rest? In addition, the ion content cannot be derived from the XRF analysis. Only the oxide composition can be determined by this analysis. Correct it.

Author reply: The remaining substances in Table 1 are acid incompatibles and the test results in Table 1 have been corrected.

Comment 6:All figures: change the shape of the marker. Circle and hexagon are not clearly distinguished

Author reply: Thanks to the reviewer's reminding, the symbols in the map have been changed by XRD.

Comment 7: Bottom part of page 5: The λ value is probably wrong. Check it carefully. Also, move the sentences about the equipment and methods used for XRD analysis to Section 2.1

Author reply: Thanks for your helpful remind, the author has made relevant revisions in the original manuscript.

Comment 8: Figure 4: As a result of TG analysis, dehydration of Mg(OH)2 was almost completed at 400°C. Nevertheless, why did the authors set the firing temperature between 400-800°C as the main test variable?

Author reply: The purpose of choosing 400-800 is not only to reduce the calcination temperature, so as to reduce energy consumption, but also to obtain a better mechanical properties.

Comment 9: Section 2.3: Add a new table showing mix proportions for better understanding.

Author reply: Section 2.3 has given a detailed description of the mixing proportions of the specimen in the original manuscript.

Comment 10: Top of page 7: The sentences describing the procedures of XRD and SEM analyses should be moved to where they first appear.

Author reply: Thanks for your helpful remind, the author has made relevant revisions in the manuscript.

Comment 11: Bottom of page 7: How was the crystal size determined?

Author reply: The full spectrum fitting analysis of the XRD results of magnesium oxide containing boron after heat treatment at different temperatures was carried out by using TOPAS software, and the grain size of different substances at different heat treatment temperatures was calculated by combining with Debye-Scherrer formula.

Debye-Scherrer: D = Kλ/Bcosθ

Comment 12: Figure 5: As confirmed by the TG analysis, the dehydration of Mg(OH)2 was almost completed at 400°C. Nevertheless, why was the Mg(OH)2 peak detected in the 400°C sample?

Author reply: Since Mg(OH)2 was not completely decomposed at 400℃, the Topas4.2 quantitative analysis results were shown in Table 4. At 400℃, there was still 15.48% Mg(OH)2, so there was still a peak of Mg(OH)2 in the XRD test results.

Comment 13: Table 5: The BET surface area of the 800°C sample is remarkably low. Didn't the authors consider the possibility of melting as a cause?

Author reply: The melting point of MgO is 2800℃, and it will not melt at 800℃. As shown in Table 5, MgO content in samples at 700℃ is 83.10%, while that at 800℃ is 82.89%, with a difference of 0.21%. The decrease of sample content may be due to the error of the instrument test process.

Comment 14:Middle part of page 10: I don't agree with the sentence “This is associated with the degree of crystallization”, because no evidence is confirmed in the XRD patterns. If there was a change in the degree of crystallization, the change from the amorphous hump to the crystallization peak should have been confirmed with increasing temperature. Based on the SEM images and BET measurement results, I would like to suggest the reduction of the specific surface area as a more probable cause regardless of the change in the degree of crystallization. Consider this, and if the authors agree with me, revise the relevant sentences. Page 11: "As mentioned" is better than "As we all know".

Author reply: Thanks for your helpful remind, the author has made relevant revisions in the manuscript.